# Fast Video Generation with SLIDING TILE ATTENTION

Peiyuan Zhang [1]  Yongqi Chen [*2]  Runlong Su [*1]  Hangliang Ding [3]
Ion Stoica [4]  Zhengzhong Liu [5]  Hao Zhang [1]

## Abstract

Diffusion Transformers (DiTs) with 3D full attention power state-of-the-art video generation, but suffer from prohibitive compute cost – when generating just a 5-second 720P video, attention alone takes 800 out of 945 seconds of total inference time. This paper introduces sliding tile attention (STA) to address this challenge. STA leverages the observation that attention scores in pretrained video diffusion models predominantly concentrate within localized 3D windows. By sliding and attending over the local spatial-temporal region, STA eliminates redundancy from full attention. Unlike traditional token-wise sliding window attention (SWA), STA operates tile-by-tile with a novel hardware-aware sliding window design, preserving expressiveness while being *hardware-efficient*. With careful kernel-level optimizations, STA offers the first efficient 2D/3D sliding-window-like attention implementation, achieving 58.79% MFU. Precisely, STA accelerates attention by 2.8–17× over FlashAttention-2 (FA2) and 1.6–10× over FlashAttention-3 (FA3). On the leading video DiT, HunyuanVideo, STA reduces end-to-end latency from 945s (FA3) to 501s *without* quality degradation, requiring no training. Enabling finetuning further lowers latency to 268s with only a 0.09% drop on VBench. We make our codebase public at https://github.com/hao-ai-lab/FastVideo.

## 1. Introduction

Diffusion Transformers (DiTs) have emerged as the leading architecture for high-resolution video generation, capable of

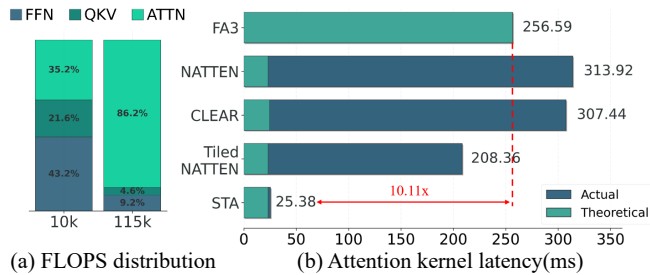

(a) FLOPS distribution     (b) Attention kernel latency(ms)

*Figure 1.* (a) Generating a 5s 720P clip in Hunyuan involves processing 115K tokens, making attention the dominant cost. (b) Attention latency comparison: existing methods fail to translate FLOP reduction into wall-clock speedup; STA is hardware-efficient and achieves proportional speedup with sparsity.

synthesizing long-duration, visually coherent outputs (Peebles & Xie, 2023; OpenAI, 2024). Central to their success is 3D attention mechanism, which models spatial and temporal dependencies by flattening video frames as a unified sequence of visual tokens (Yang et al., 2024b; Genmo-Team, 2024; HunyuanVideo-Team, 2025). However, this design introduces significant computational overhead due to the quadratic complexity of attention, making training and inference prohibitively slow as video resolutions and durations increase. As illustrated in Figure 1, attention computation dominates the overall inference cost. Even with FlashAttention 3 (FA3) (Shah et al., 2024) and a high-end H100 GPU, HunyuanVideo (HunyuanVideo-Team, 2025) still requires *16 minutes* to generate a 5-seconds 720p video. This bottleneck severely limits the practical deployment of DiTs.

Video data inherently exhibit high redundancy – adjacent frames exhibit minimal differences, and spatially close pixels tend to have stronger correlations. This redundancy suggests that treating every token independently in 3D attention may be unnecessarily expensive. In this paper, we hypothesize that such redundancies are carried by 3D full attention in pretrained video diffusion models, which, if properly exploited, can drastically accelerate inference. To verify this, we visualize the attention scores of HunyuanVideo in Figure 2. The results reveal an intriguing *3D locality* pattern: queries assign significantly higher attention scores to spatially and temporally nearby keys. To quantify this effect, we compute attention recall, measuring the fraction of total

---

[*]Co-second authorship  [1]University of California, San Diego
[2]University of Michigan, Ann Arbor  [3]Tsinghua University
[4]University of California, Berkeley  [5]Mohamed bin Zayed University of Artificial Intelligence. Correspondence to: Hao Zhang <haozhang@ucsd.edu>.

*Proceedings of the 42^{nd} International Conference on Machine Learning*, Vancouver, Canada. PMLR 267, 2025. Copyright 2025 by the author(s).

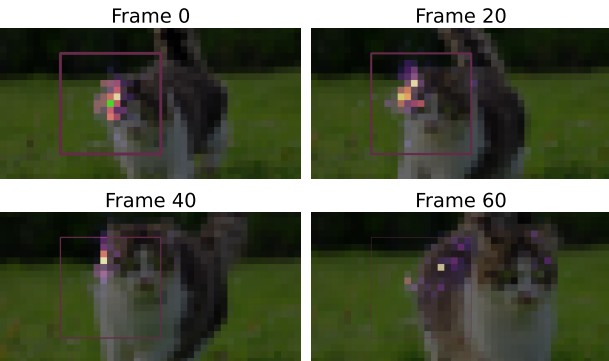

*Figure 2.* Visualization of attention locality. The green point means the query point and the magma-colored regions indicate areas of high attention values in response to the query. Instead of attending to the entire image, the query's attention forms a concentrated local hotspot.

attention scores concentrated within a local window. As shown in Figure 3 left, despite training with full 3D attention, HunyuanVideo exhibits strong locality: on average, a local window covering only 15.52% of the total token space accounts for 70% of the total attention score.

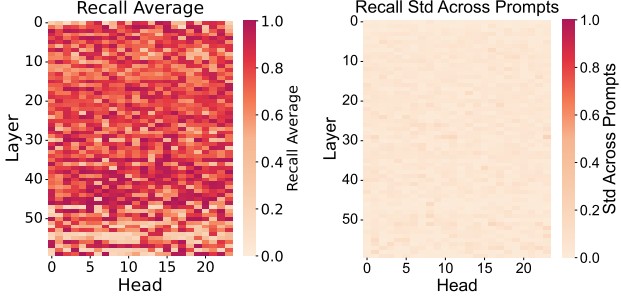

*Figure 3. Left*: Fraction of attention scores within a (12, 24, 24) local window across diffusion steps and 10 different prompts. Most heads show high recall, indicating a local attention pattern. *Right*: Despite the different recall across heads, the standard deviation across prompts remains low.

This observation seemingly suggests that sliding window attention (SWA) is an ideal alternative to retain attention expressiveness while reducing computational cost. However, existing 2D or 3D SWA implementations, such as NATTEN (Hassani et al., 2023) and CLEAR (Liu et al., 2024), fail to translate FLOP reductions into proportional wall-clock speedups, as shown in Figure 1b. Their inefficiency arises because higher-order (2D/3D) sliding window attention creates a highly irregular attention mask, wasting many computations and generating significant masking overhead. The computation pattern for SWA is inherently *GPU-unfriendly*, resulting in poor hardware utilization (see §2.2).

To overcome this, we develop SLIDING TILE ATTENTION (STA), a hardware-aware attention mechanism that rethinks sliding window computation via system-algorithm co-design. We define a tile as a contiguous group of tokens forming a spatial-temporal cube, with its size determined by the block size in FlashAttention.Instead of sliding over contiguous tokens, STA operates tile-by-tile, enabling efficient memory access and parallelism while effectively preserving the 3D locality. Inspired by FA3, STA adopts a consumer-producer paradigm, where producer warpgroups *asynchronously* load data from HBM to SRAM while consumer warpgroups compute attention. Because STA slides over tiles instead of individual tokens, it eliminates the need for explicit attention masking at computation, a significant overhead observed in other SWA implementations. The sparse attention mask is managed entirely by the producer warpgroups; hence, the computation on consumer wrap groups remains dense and hardware-efficient. As a result, STA is the first higher-order sliding-window-like attention to achieve wallcock speedups proportional to sparsity (Figure 1 (b)).

Besides efficient computation, selecting optimal window sizes is crucial to preserving generation quality. We find that different attention heads exhibit specialized locality patterns – some heads focus on finer details in a small area, yet others capture broader context at a larger window – which we term as *head specialization*. Importantly, this head specialization remains agonistic to prompts, as evidenced in Figure 3. Based on this property, we develop a simple yet effective method to automatically configure the optimal window size *per head* via profiling, striking a balance between efficiency and quality. With STA, HunyuanVideo can generate a 5-second 720P video in 501s with no or minimal quality loss in a plug-and-play manner. In comparison, HunyuanVideo with FlashAttention-2 takes 1496s, while FlashAttention-3 takes 945s. STA achieves a end-to-end speedup of 2.98× over FlashAttention-2 and 1.89× over FlashAttention-3. Additionally, by fine-tuning diffusion models under more radical attention sparsity, we unlock even greater efficiency, delivering a 2.43 - 3.53× end-to-end speedup compared with FlashAttention-3.

This paper makes the following contributions: (1) We identify and quantify *3D locality* and *head specialization* in state-of-the-art video DiTs, revealing substantial redundancy in full 3D attention. (2) We introduce SLIDING TILE ATTENTION, a tile-based sliding window attention mechanism. Our optimized kernel achieves minimum overhead compared to FlashAttention 3 with an MFU of 58.79%. (3) STA accelerates attention by > 10× and end-to-end video generation by up to 3.53× with no or minimum quality loss.

# 2. Problem

In this section, we provide background on why sliding window attention (SWA) is inefficient in high-dimensional settings, particularly for Video Diffusion Transformers (Peebles & Xie, 2023). For clarity, our notation assumes cubic window, tile, and video sizes unless stated otherwise, though our approach extends to non-cubic configurations.

## 2.1. Attention in Video DiTs

State-of-the-art Video DiTs employ 3D full attention to mix signals across tokens, allowing each token to attend to any other token. Given a video latent of shape $(L, L, L)$ (often encoded via a VAE), this is achieved by flattening the 3D data into a sequence of length $L^3$ and applying full bidirectional attention. However, as the sequence length grows cubically with $L$, even a small increase in resolution or duration leads to a significant computational burden. As a result, applying 3D attention to high-resolution, long-duration videos becomes prohibitively expensive.

Formally, let $\mathbf{Q}, \mathbf{K}, \mathbf{V} \in \mathbb{R}^{N \times d}$ represent the query, key, and value of input sequences for a single attention head, where $N = L^3$, and $d$ is the dimension of each head. Let $M \in \{-\infty, 0\}^{N \times N}$ represents the attention mask. The attention operation is defined as:

$$S = \frac{QK^\top}{\sqrt{d_k}}, \quad A = \text{Softmax}(S + M), \quad O = AV \quad (1)$$

A naive attention implementation constructs the full $S, A, M \in \mathbb{R}^{N \times N}$ on GPU HBM, leading to both $\mathcal{O}(N^2)$ memory overhead and excessive data transfers between HBM and SRAM. FlashAttention mitigates this issue by tiling input sequences into smaller blocks. Through online softmax, each GPU SM loads a block of queries, keys, and values into SRAM, performs computation, and writes the final result to HBM, avoiding the materialization of $A$ and $S$. To reduce computation cost, we can apply attention mask to control sparsity, with the mask computed on-chip per block to avoid the $\mathcal{O}(N^2)$ memory cost of a global mask. This sparsity can reduce latency by skipping masked-out attention regions. However, as we show in the next section, this is not always effective.

Sliding window attention (SWA) is a widely used sparse attention method that reduces computation costs while preserving locality. In SWA, a query attends only to keys within a fixed window, and stacking multiple attention layers naturally expands the receptive field beyond the window size. SWA has been extensively studied in natural language processing (Beltagy et al., 2020; Jiang et al., 2023). As motivated in §1, state-of-the-art video diffusion models exhibit a strong *3D locality* pattern, making them a natural candidate for applying 3D SWA. However, directly applying SWA to high-dimensional data fails to fully utilize GPU computation. Existing 2D/3D SWA kernels, such as Tiled NATTEN (Hassani et al., 2023), shift window centers at image and video boundaries to ensure each query attends to a constant number of keys. It also improves kernel efficiency through input tiling and kernel fusion, but its performance suffers from problems described next.

## 2.2. Inefficiency of Sliding Window Attention

Implementing 2D/3D SWA with FlashAttention requires defining its attention mask. As discussed in §2.1, FlashAttention calculates and applies masks at the block level. Based on different intra-block masking strategies, we categorize attention blocks into three types: dense (with all attention scores retained), empty (mask out all values), and mixed (with some scores removed), shown in Figure 4. Empty blocks can be entirely skipped during computation. Mixed blocks, while sparser than dense blocks, introduce significant overhead. First, they require full computation for the entire block before applying masks to retain or discard attention scores, introducing unnecessary computations. Second, to determine which position to retain or discard, the attention kernel needs to calculate the value of the mask based on the SWA pattern and the block's position relative to the entire attention mask. The mask calculation introduces substantial overhead – in the case of calculating the simple causal mask, FlexAttention (Dong et al., 2024) reports a 15% overhead. We show in §4.1 that the mask overhead dominates the attention latency for more complex masking patterns such as sliding windows. In SWA, each query attends to a distinct set of keys, resulting in a zigzag pattern in the attention map and generating numerous mixed blocks, as shown in Figure 4 (a). Although the state-of-the-art sliding window attention implementation, Tiled NATTEN, aims to reorder the inputs to increase the number of dense blocks, a significant portion of blocks remains as mixed blocks.

In summary, SWA beyond 1D sequences introduces two major inefficiencies: (1) mixed blocks do not reduce FLOPs due to sparsity, and (2) they incur additional mask evaluation overhead, making them slower than dense blocks. Efficient sparse attention in 3D should minimize mixed blocks while maintaining locality. This insight drives the development of our novel STA method, which significantly reduces mixed blocks while preserving the locality property of SWA.

## 2.3. Alternative Methods for 3D Attention

Other than sparsifying the attention maps, some approaches accelerate video diffusion by decomposing the 3D attention into alternating spatial and temporal components (Singer et al., 2022; Wang et al., 2023; Chen et al., 2023; Ma et al., 2024). However, these methods have been largely superseded by full 3D attention in state-of-the-art video mod-

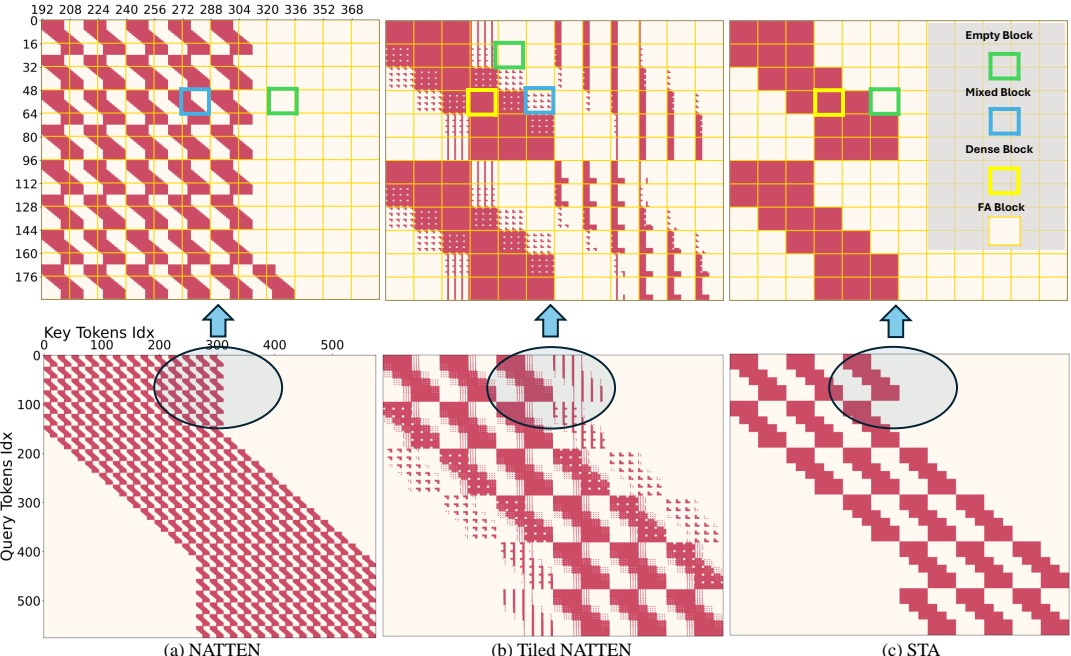

*Figure 4.* The attention map of NATTEN, Tiled NATTEN, and STA. We plot with an image size 24×24 and a 12×12 local window. The tile size is set to 4×4. (a) NATTEN creates many mixed blocks that are very inefficient for Flash Attention computation. (b) Tiled NATTEN increases the number of dense blocks, but the mixed blocks persist. (c) STA completely eliminates the mixed block, making the computation extremely friendly for GPU. Note that we mainly show STA's application in 3D scenarios for video generation in this paper, but for better illustration, we present the 2D scenario in this plot.

els (Zheng et al., 2024; Lin et al., 2024; Genmo-Team, 2024; HunyuanVideo-Team, 2025). We hypothesize that this is because alternating spatial and temporal attention fails to capture interactions between tokens that are offset in both spatial and temporal dimensions. For instance, a query at (1, 1, 1) can attend to keys at (1, X, X) or (X, 1, 1), but not at (2, 2, 2), even though they are spatially close. This disrupts the *3D locality* pattern, which we have shown as a key characteristic of video diffusion models.

## 3. Methods

### 3.1. SLIDING TILE ATTENTION

In vanilla sliding window attention, each query attends to a local window centered around it, resulting in different queries attending to distinct key groups. This lack of shared attention key groups is the root cause of irregularities in SWA's attention map, creating mixed blocks. We propose SLIDING TILE ATTENTION (STA), a novel sliding window attention variant that exclusively operates on dense blocks and empty blocks. As shown in Figure 5, STA organizes queries and keys into tiles. All queries in the same tile attend to the same set of keys within their common local window, ensuring a more structured attention pattern. By setting the tile area equal to the block size in FlashAttention and

*Table 1.* Ratio of dense and mixed blocks for tiled NATTEN and STA with tile size (4,4,4) and video size (48,48,48). STA generate only dense blocks, which is more computationally friendly than mixed blocks on GPU.

| Attention | Window Size | Dense Block | Mixed Block |
|---|---|---|---|
| Tiled NATTEN | (11,11,11) | 0.06% | 7.17% |
| STA | (12, 12, 12) | 1.56% | 0.0% |
| STA | (20, 20, 20) | 7.23% | 0.0% |

arranging queries in a tile with consecutive token indices, we form dense FlashAttention blocks. This design allows each query tile to attend densely to key tiles within the window, eliminating mixed blocks and improving compute efficiency. We illustrate STA attention mask in Figure 4 (c).

Formally, for 3D STA, given a video of dimension $(L, L, L)$ and a FlashAttention block size of $(B, B)$, STA sets the tile size $T$ such that $B = T^3$. It further assumes that both the video size $L$ and window size $W$ are integer multiples of $T$. The video is partitioned into non-overlapping tiles of size $(T, T, T)$, and flattened into 1D sequence in a way that tokens within the same tile have consecutive sequence indices. Conceptually, STA slides the window with a step size of $(T, T, T)$. For each step, it computes attention between the central query tiles and all key tiles within the window, producing $\left(\frac{W}{T}\right)^3$ dense attention blocks without

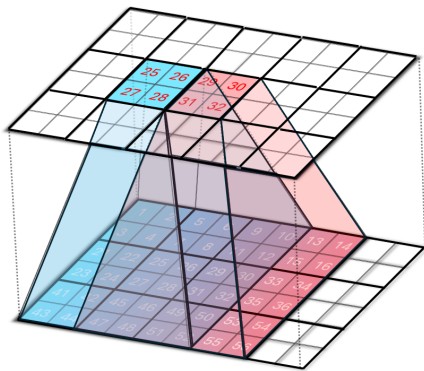

*Figure 5.* 2D SLIDING TILE ATTENTION with tile size (2, 2) and window size (6, 6). After attending to all the key tiles, each query tile will generate nine 4x4 dense blocks in the attention map. We showcase 2D STA for better illustration. 3D STA can be inferred similarly.

mixed blocks.

To demonstrate STA superiority in creating a *GPU-friendly* compute pattern, we give the following formula to quantitatively measure the different types of blocks in 3D Tiled NATTEN and 3D STA.

**Theorem 3.1.** *Consider a tiled NATTEN configuration with tile size* $(T, T, T)$, *window size* $(W, W, W)$, *and video size* $(L, L, L)$. *Let the FA block size be* $(B, B)$, *where* $B = T^3$. *Ignoring boundary effects, the number of dense blocks is given by:*

$$N_{\text{dense}} = \left( \max\left( 2 \left\lfloor \frac{W+1}{2T} \right\rfloor - 1, 0 \right) \right)^3 \cdot \left( \frac{L}{T} \right)^3.$$

*The number of mixed blocks in tiled NATTEN is:*

$$N_{\text{mix}} = \left( 2 \left\lceil \frac{W-1}{2T} \right\rceil + 1 \right)^3 \cdot \left( \frac{L}{T} \right)^3 - N_{\text{dense}}.$$

Intuitively, for a block to be dense in NATTEN, the window size should be at least twice the size of the tile size, such that the left-most query in the tile can attend to the right-most query. On the other hand, the left-most query in a tile can still attend to keys that are $\frac{W-1}{2T}$ tiles further left, creating mixed blocks.

**Theorem 3.2.** *With the same notation, if* $W$ *is an integer multiple of* $T$, *the number of dense blocks in* SLIDING TILE ATTENTION *is:*

$$S_{\text{dense}} = \left( \frac{W}{T} \right)^3 \cdot \left( \frac{L}{T} \right)^3.$$

*All remaining blocks are empty and there are no mixed blocks.*

Intuitively, each query tile will only attend to its local window in STA, which has $\left( \frac{W}{T} \right)^3$ tiles of keys, creating the same number of blocks in the attention map. We apply Theorem 3.1 and Theorem 3.2 to calculate the ratio of different blocks and report them in Table 1.

**Kernel-level optimization.** STA can be efficiently implemented with FlexAttention, which provides enough functionality to skip all empty blocks and avoid adding unnecessary intra-block masks on the dense blocks. We can further optimize the sparse attention masks by disaggregating the inter-block mask logic from the compute kernels. Thus, we implement our attention kernels based on ThunderKittens (Spector et al., 2024) and FlashAttention3 (Shah et al., 2024). Our implementation splits the threadblock into compute warpgroups and data warpgroups, and the inter-block mask is completely managed by the data warpgroups. Each compute warpgroup is responsible for calculating one query block, which always resides in the SRAM (Split-Q (Dao, 2024)). The data warpgroup is responsible for asynchronously loading the KV blocks from HBM to SRAM. For each block of query, the data warpgroup needs to decide which key and value blocks the query block will attend to in STA and only load those blocks. Since the data warpgroups are asynchronous, the overhead of calculating the inter-block mask in STA and deciding which data to load can be hidden with overlapping. On the other hand, the compute worker is completely oblivious of the sparse attention pattern. It performs attention computation with the key value blocks in shared memory loaded by data workers, and once all data is consumed in the circular cache, the computation is finished.

### 3.2. Applying STA to Video Diffusion Model

We can either apply STA to directly replace the 3D attention in pretrained video DiTs without training, or with small amount of training which enables even greater sparsity.

**Training-free.** As illustrated in Fig. 3, video diffusion models exhibit a pronounced *3D locality* and *head specialization* pattern. Different transformer heads have different levels of locality, but their pattern is largely consistent across different prompts. We can exploit this property to search for the optimal window size for each head on a very small number of prompts, and expect the search result to work well on other prompts. We develop a simple heuristics to find such configuration in Algorithm 1 and decide the final configuration by averaging the mask-search loss across 16 prompts. In practice, we keep full attention for the initial $T_0$ timesteps (Li et al., 2024a; Zhao et al., 2024; Lv et al., 2024), and then apply STA for the rest timesteps.

**Finetuning.** Beyond searching for the optimal mask per attention head without tuning, we can fix a window size with a high sparsity and fine-tune the model to adapt. Since

HunyuanVideo – 15 mins 45 s

STA-tf-1.89x – 8 mins 21 s

STA-t-2.43x – 6 mins 29 s

Δ-DiT-1.36x – 11 mins 34 s

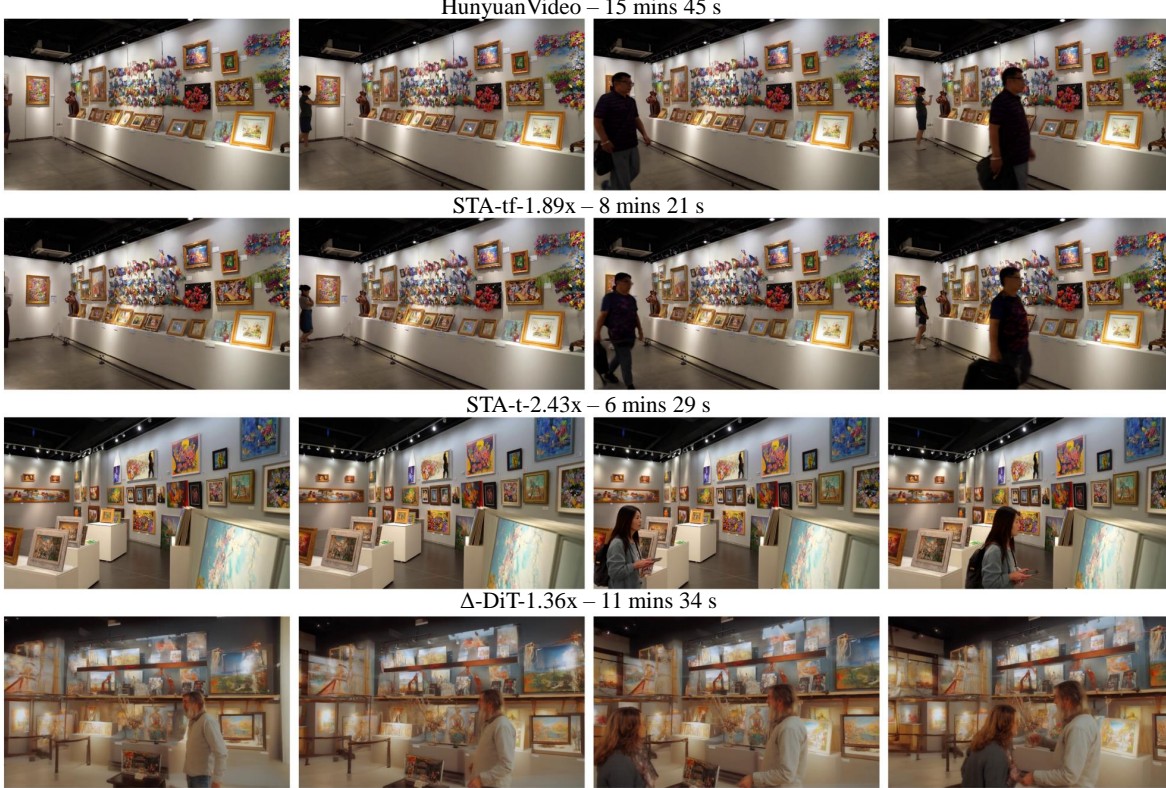

Prompt: Tour of an art gallery with many beautiful works of art in different styles.

*Figure 6.* Qualitative example of 720P 5-second videos. While fine-tuning introduces minor shifts in the output distribution of STA-t-2.43x, the model still preserves high video generation quality. Videos generated by Δ-DiT are generally less sharp than those generated by the original HunyuanVideo and STA.

---

**Algorithm 1** STA Mask Search

---

**Input:** Transformer model $M$, Total steps $T$, Mask pattern
list $\mathcal{P}$, Keep first $T_0$ timestep full attn
**Output:** Dictionary dict that stores selected mask pattern
for each head
  Initialize dict
  **for** $t = T_0 + 1$ to $T$ **do**
    **for** each layer head combination $(l, h)$ in $M$ **do**
      $O \leftarrow$ (attn output of original $(l, h)$ )
      Initialize minimum_loss $\leftarrow \infty$
      Initialize best_pattern $\leftarrow$ null
      **for** each $p$ in $\mathcal{P}$ **do**
        Mask head $h$ for layer $l$ using mask pattern $p$
        $O' \leftarrow$ (attn output of $M$ after masking)
        loss $\leftarrow MSE(O, O')$
        **if** loss < minimum_loss **then**
          minimum_loss $\leftarrow$ loss
          best_pattern $\leftarrow p$
      Record best_pattern for $(t, l, h)$ in dict
  **return** dict

---

STA follows the *3D locality* property, this adaptation can be learned efficiently with minimal training overhead (in our experiments, 8 hours on 8 H100, which is minimal compared to the pretrain cost of video diffusion models). Although each attention layer is restricted to a local window, the receptive field expands through stacked transformer layers, enabling the Diffusion Transformer to generate globally coherent videos in the end.

We use three different loss terms during finetuning. The attention distillation loss directly supervises the intermediate attention patterns of our STA to match the original dense attention behaviors:

$$\mathcal{L}_{attn} = \frac{1}{N} \sum_{i=1}^{N} \|f_\phi^{(i)}(x_t, t, c) - f_\psi^{(i)}(x_t, t, c)\|_2^2, \quad (2)$$

where $f_\phi^{(i)}$ and $f_\psi^{(i)}$ denote the intermediate attention outputs from the $i$-th transformer layer of our sliding tile model and the original attention teacher. This loss ensures each sparse attention layer to approximate its corresponding dense attention teacher. We also add a final layer loss to align the final output of the student and teacher:

$$\mathcal{L}_{final} = \|f_\phi(x_t, t, c) - f_\psi(x_t, t, c)\|_2^2 \quad (3)$$

Additionally, we employ a data loss following the flow matching formulation (Esser et al., 2024; Lipman et al., 2022):

$$\mathcal{L}_{data} = \|(f - x_0) - f_\phi(x_t, t, c)\|_2^2, \quad (4)$$

where $x_0$ represents the VAE latent of the input frame, $x_t$ is the noised latent at diffusion step $t$, and $c$ denotes the text embedding.

The complete objective combines these terms:

$$\min_\phi \mathbb{E}_{x \sim p(x), c \sim N(0,1), t}[\alpha\mathcal{L}_{data} + \beta\mathcal{L}_{final} + \gamma\mathcal{L}_{attn}] \quad (5)$$

The detailed training setup can be found in Appendix B.

## 4. Experiments

We evaluate STA on HunyuanVideo, a state-of-the-art open video DiT comparable to many proprietary ones[1]. We generate Hunyuan outputs with 117 frames at a 1280×768 resolution. After VAE compression and tokenization, this corresponds to a latent video of shape (30, 48, 80). Beyond video, we also apply STA on the leading image diffusion model, FLUX (Black-Forest, 2023), to demonstrate its effectiveness in 2D. We evaluate both *efficiency* and *video quality*. STA kernel's efficiency is measured using standard metrics such as MFU and latency, as detailed in §4.1. For end-to-end speedup on DiT, we report measured wall-clock latency, excluding time spent on VAE and text encoder. For generated video quality, we find existing automated metrics are often unreliable. Following Polyak et al. (2024), we emphasize human evaluation and present the results in §4.2. For completeness, we also report automated metrics, including VBench (Huang et al., 2024), SSIM, PSNR, and CD-FVD (Ge et al., 2024). We provide an example in Figure 6, with additional qualitative results available in Appendix Section G.

**Baseline methods.** We compare STA to other sparse or window attention designed for image or video, including CLEAR (Liu et al., 2024), NATTEN (Hassani et al., 2023), and Swin (Liu et al., 2021b). Also, we adapt the caching-based method, Δ-DiT (Chen et al., 2024), for evaluation on HunyuanVideo. Further details on the baseline methods and their implementations can be found in Appendix C.

### 4.1. Efficiency of SLIDING TILE ATTENTION

We benchmark the efficiency of various attention algorithms assuming generating 720P 5s videos using Hunyuan, shown in Table 2. The configuration ensures that all sparse kernels maintain approximately 90% sparsity, with additional

---

[1]We skip evaluating on other open models (Lin et al., 2024; Zheng et al., 2024; Ma et al., 2024) due to their significantly lower overall quality compared to Hunyuan.

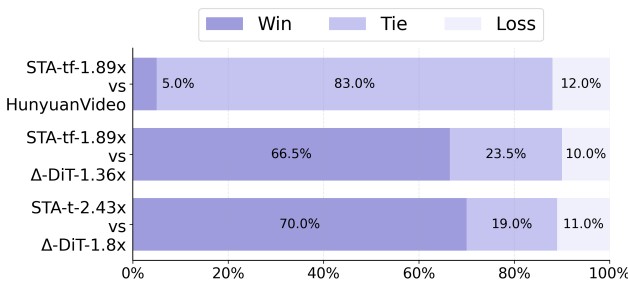

*Figure 7.* Human evaluation on 200 prompts from the MovieGen Bench (Polyak et al., 2024). STA achieves a 1.89× end-to-end speedup while maintaining performance comparable to the original HunyuanVideo. Additionally, STA consistently outperforms Δ-DiT across different inference budgets.

results for a lower sparsity setting (56%) provided in table 7. Since STA builds on FA3 and ThunderKittens, we use ThunderKittens' FA3 as the baseline and report the relative speedup of all sparse attention kernels. To quantify efficiency, We introduce *kernel efficiency*, defined as the ratio of a sparse kernel's MFU to that of full attention. This metric captures how well sparse kernels translate theoretical FLOP reductions into actual latency improvements.

The results highlight the inefficiency of existing methods. Despite reducing TFLOPs to 15.65, CLEAR incurs a 0.86× slowdown. Similarly, NATTEN variants, despite reaching 0.91 sparsity, still suffers from inefficiency: its vanilla version slows down by 0.85×, while its optimized tiled variant in FlexAttention achieves only a modest 1.27× speedup. Among existing methods, Swin (Liu et al., 2021a) is the only kernel with MFU exceeding 40% and kernel efficiency above 60%. However, Swin is not a sliding-window-based attention, and we argue its efficiency comes at the cost of expressiveness in §4.4.

Compared to Tiled NATTEN, one of the most optimized sliding window attention implementations, the key algorithmic difference in SLIDING TILE ATTENTION is changing the sliding unit from a token to a tile. Despite its simplicity, this modification significantly improves efficiency. To ensure a direct comparison with tiled NATTEN, we also implement STA in FlexAttention – STA improves MFU from 8.20% to 41.03%. Further, with our optimized kernel for asynchronous data loading and inter-block mask management in ThunderKittens, STA achieves a 10.45× speedup over full attention. Additionally, we evaluate STA with 58.33% sparsity, where it achieves 2.37x speedup. This efficiency gain enables a significantly larger window size while still outperforming NATTEN. To our knowledge, STA is the first sliding-window sparse attention that achieves both 3D locality and hardware efficiency.

*Table 2.* Forward speed of sparse attention kernels in a setup aligned with HunyuanVideo's inference configuration (bf16, 720P, 5s, 115.2K seq_len, $d_{head}$ = 128, # heads = 24). Config controls the window size of each sparse attention.

| Methods | Implementation | Config | Sparsity | TFLOPS | Latency(ms) | MFU | Kernel Efficiency | Speedup |
|---------|----------------|--------|----------|--------|-------------|-----|-------------------|---------|
| FA 3 | ThunderKittens | - | 0.00% | 164.03 | 265.28 | 62.49% | 100.00% | 1.00× |
| FA 3 | CUDA | - | 0.00% | 164.03 | 256.59 | 64.61% | 103.39% | 1.03× |
| CLEAR | FlexAttention | r=16 | 90.46% | 15.65 | 307.44 | 5.15% | 8.24% | 0.86× |
| NATTEN | FlexAttention | w=(19,25,25) | 89.69% | 16.91 | 313.92 | 5.44% | 8.71% | 0.85× |
| Tiled NATTEN | CUDA | w=(19,25,25) | 89.69% | 16.91 | 458.36 | 3.73% | 5.97% | 0.58× |
| Tiled NATTEN | FlexAttention | w=(19,25,25) | 89.69% | 16.91 | 208.36 | 8.20% | 13.12% | 1.27× |
| Swin | FlexAttention | w=(24,32,32) | 87.42% | 20.64 | 47.90 | 43.55% | 69.69% | 5.54× |
| STA | FlexAttention | w=(18,24,24) | 91.00% | 14.76 | 36.36 | 41.03% | 65.66% | 7.30× |
| STA | ThunderKittens | w=(30,40,40) | 58.33% | 68.35 | 111.73 | **61.82%** | **98.93%** | 2.37× |
| STA | ThunderKittens | w=(18,24,24) | 91.00% | 14.76 | **25.38** | 58.79% | 94.09% | **10.45×** |

## 4.2. Human Evaluations

We assess human preference across five models that achieve the best quality performance: (1) *HunyuanVideo*; (2) STA-*tf-1.89x*: HunyuanVideo with 1.89× speedup via training-free mask search, (3) STA-*t-2.43x*: HunyuanVideo with 2.43× speedup via finetuning with STA, (4-5) two variants of Δ-DiT (1.36× and 1.8× speedup). Other baselines such as CLEAR or Swin are either prohibitively slow or produce subpar quality. Following MovieGen (Polyak et al., 2024), we randomly sample 200 prompts from the MovieGen Bench and conduct pairwise comparisons between these models. Evaluators select the video with higher overall quality or mark both as a tie.

In Figure 7, STA-t-2.43x decisively outperforms Δ-DiT-1.8x, achieving a dominant 70.0% win rate versus 11.0%, despite a greater speedup. Similarly, STA-tf-1.89x surpasses Δ-DiT-1.36x with a 66.5% win rate against 10.0%. Compared to the original HunyuanVideo, STA maintains competitive quality, with STA-tf-1.89x achieving a 83.0% tie rate—indicating near-parity in most cases. Though it has a 7.0 percentage point lower win rate than its loss rate, this tradeoff comes with a 1.89× speedup, demonstrating strong quality preservation alongside significant efficiency gains. These results establish STA as achieving a superior quality-efficiency tradeoff compared to Δ-DiT.

## 4.3. Training-free Results

In Table 3, we evaluate mask-search STA and Δ-DiT on VBench prompts, testing robustness across different sampling steps. For each diffusion step count, we report SSIM, PSNR, and CD-FVD, using HunyuanVideo's outputs at the same step count as the reference. We set the scheduler shift to 17 for 10-step inference and 7 for 25- and 50-step inference following Hunyuan's default settings. We keep full attn for the first 12,6,3 steps for 50-step, 25-step, and 10-step

*Table 3.* Training-free performance with varying sampling steps. Δ-DiT shows consistently worse quality compared to STA

| Model | SSIM ↑ | PSNR ↑ | CD-FVD ↓ | Latency | Speedup |
|-------|--------|--------|----------|---------|---------|
| *steps = 50* | | | | | |
| Δ-DiT | 72.86 | 18.09 | 122.74 | 693s | 1.36× |
| STA | 87.67 | 28.76 | 66.12 | 501s | 1.89× |
| *steps = 25* | | | | | |
| Δ-DiT | 77.91 | 19.86 | 196.25 | 352s | 1.34× |
| STA | 88.96 | 28.99 | 76.34 | 250s | 1.89× |
| *steps = 10* | | | | | |
| Δ-DiT | 83.19 | 21.20 | 201.24 | 144s | 1.32× |
| STA | 87.84 | 27.14 | 84.80 | 105s | 1.76× |

inference respectively.

Our training-free STA consistently outperforms Δ-DiT even with much higher speedup. At 50 steps, STA achieves substantial improvements in all metrics, with 14.81 higher SSIM (87.67 vs. 72.86) and 10.67 higher PSNR (28.76 vs. 18.09). Δ-DiT's CD-FVD score is 56.62 higher than STA (122.74 vs. 66.12, where lower is better). This gap grows to 119.91 at 25 steps and 116.44 at 10 steps. Qualitatively, Δ-DiT consistently produces visually degraded outputs across all sampling steps, exhibiting compromised structural similarity and diminished fine details, while STA maintains high fidelity to the original model.

## 4.4. Finetuning Results

Fine-tuning on new data introduces slight distribution shifts, meaning the same prompt may yield different, yet high-quality, video variants. Consequently, similarity metrics like PSNR become less suitable, and we instead rely on VBench(Huang et al., 2024), a comprehensive benchmark for video generation. We first examine the impact of directly replacing full attention with sparse attention, without tuning, to evaluate how well each algorithm approx-

*Table 4.* Performance on VBench across different sparse attention patterns. STA achieves both high-quality video generation and significant speedup, while CLEAR and Tiled NATTEN suffer from efficiency issues and Swin suffers from quality degradation.

| Methods | Config | VBench Quality | VBench Semantic | VBench Total | Attn Sparsity | PFLOPS | Latency | Speedup |
|---------|--------|-----------------|------------------|---------------|----------------|--------|---------|---------|
| FA2 | – | 85.34% | 72.17% | 82.71% | 0.00% | 574.16 | 1496s | 0.63× |
| FA3 | – | 85.34% | 72.17% | 82.71% | 0.00% | 574.16 | 945s | 1.00× |
| *w.o training* | | | | | | | | |
| CLEAR | r=32 | 84.41% | 74.20% | 82.37% | 56.23% | 280.90 | 2567s | 0.37× |
| Tiled NATTEN | w=(30,41,41) | 84.61% | 75.00% | 82.69% | 58.33% | 269.92 | 1858s | 0.51× |
| Swin | w=(48,64,64) | 80.91% | 71.35% | 79.00% | 55.81% | 283.11 | 762s | 1.24× |
| Swin | w=(30,40,40) | 78.84% | 72.28% | 77.53% | 76.49% | 175.20 | 497s | 1.90× |
| STA | w=(30,40,40) | 84.63% | 73.83% | 82.46% | 58.33% | 269.92 | 527s | 1.79× |
| STA | w=(18,24,24) | 81.47% | 77.03% | 80.58% | 91.00% | 99.54 | 268s | 3.53× |
| *w. training* | | | | | | | | |
| Swin | w=(30,40,40) | 77.50% | 67.39% | 75.48% | 55.81% | 283.08 | 497s | 1.90× |
| STA | w=(30,24,40) | 85.37% | 73.52% | 83.00% | 75.00% | 182.99 | 388s | 2.44× |
| STA | w=(18,24,24) | 84.76% | 74.05% | 82.62% | 91.00% | 99.54 | 268s | 3.53× |

imates full 3D attention. In Table 4, CLEAR and Tiled NATTEN retain reasonable video quality (VBench scores of 82.37% and 82.68%, respectively) compared to full attention (82.71%). However, despite sparsifying attention, these methods paradoxically increase end-to-end inference latency. Swin presents the opposite challenge: while it achieves moderate speedup (1.24×–1.90×), its rigid, non-overlapping window partitions prevent local queries and keys from attending to each other if they fall into separate windows, violating the 3D locality property. This results in degraded video quality, and crucially, fine-tuning with Swin attention not only fails to recover performance but further lowers the VBench score. In contrast, STA addresses both quality and efficiency limitations. With a window configuration of $w_t$=(3,3,3), it achieves 91.00% attention sparsity, yielding a 5.76× FLOPs reduction and a 3.53× actual latency reduction.[2] Importantly, this efficiency gain comes with minimal quality tradeoff: STA maintains an 80.58% VBench score in the training-free setting and improves to 82.62% with fine-tuning.

### 4.5. Results on Image Super-Resolution

We also apply STA to speed up image superresolution with SDEdit (Meng et al., 2022). We find that FLUX with STA achieves comparable generation quality to CLEAR while offering significantly higher efficiency. Experiments for can be found in Appendix E.

## 5. Related Work

We review additional related work in diffusion acceleration. Linear attention methods (Wang et al., 2020; Liu et al.,

2021a; Arar et al., 2022; Yang et al., 2024a) can decompose the softmax operation in quadratic attention using kernel or gate functions to achieve linear complexity. However, these methods have not yet been successful in video DiTs. Concurrent to our work, quantized attention (Zhang et al., 2024; 2025a;b) and sparse attention (Zhang et al., 2025c; Xi et al., 2025) are proposed with different design for video DiTs. Another major bottleneck in diffusion models is the large number of diffusion steps. Several techniques have been proposed to accelerate sampling without sacrificing quality, including DDIM (Song et al., 2020) and faster ODE and PDE solvers using numerical methods (Song & Ermon, 2019; Lu et al., 2022a;b; Jolicoeur-Martineau et al., 2021). New methods have also emerged to further reduce the number of sampling steps, such as consistency distillation (Kim et al., 2023; Song et al., 2023; Salimans et al., 2024; Xie et al., 2024), adversarial distillation (Sauer et al., 2023), and other distillation approaches (Li et al., 2024b; Yin et al., 2023; 2024). STA is largely complementary to these methods.

## 6. Conclusion and Future Work

We introduce SLIDING TILE ATTENTION to accelerate video diffusion models, with an optimized kernel for high-order sliding-window-like attention, enabling efficient GPU execution while preserving the locality property. Experiments demonstrate that SLIDING TILE ATTENTION accelerates video generation with minimal or no quality loss. Conceptually, our method is orthogonal to other acceleration techniques, such as caching and consistency distillation. We plan to explore their combined effectiveness for further efficiency gains in future work.

---

[2]Other memory-bound operations, such as LayerNorm and modulation, likely contribute to inference overhead, preventing the full FLOPs reduction from translating directly into speedup.

## Acknowledgements

We would like to thank Will Lin and Wei Zhou for their open-source contribution in FastVideo. The work is supported by UCSD HDSI, Nvidia, and a faculty research award from Google.

## Impact Statement

Our work addresses computational bottlenecks in Diffusion Transformers by introducing efficient attention kernels that reduce video generation time while maintaining output quality. The improved efficiency makes video generation more practical for researchers and developers working with limited computing resources, potentially benefiting AI-driven video applications across creative industries, education, and so on. While faster video generation could potentially enable misuse, existing content detection and watermarking techniques can help mitigate such risks.

Overall, the benefits of more efficient video generation significantly outweigh potential concerns, representing a meaningful step toward accessible video AI systems.

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

## A. Further Details of SLIDING TILE ATTENTION

**Mask of 3D NATTEN** Algorithm 2 defines the attention mask in 3D NATTEN. First, it computes the window center for each query token. If the query is near the video edges, the center shifts inward to stay within bounds. Next, it determines the query's attention window within the spatiotemporal neighborhood. Finally, the mask is constructed by enforcing spatiotemporal constraints on query-key distances.

---

**Algorithm 2** Mask Definition of 3D NATTEN

---

**Input:** Query coordinates $(q_t, q_h, q_w)$, Key coordinates $(k_t, k_h, k_w)$, Video size $(L_t, L_h, L_w)$, Window size $(W_t, W_h, W_w)$

**Compute window center:**

$q_{ct} \leftarrow \max\left(\min\left(q_t, L_t - 1 - \frac{W_t}{2}\right), \frac{W_t}{2}\right)$

$q_{ch} \leftarrow \max\left(\min\left(q_h, L_h - 1 - \frac{W_h}{2}\right), \frac{W_h}{2}\right)$

$q_{cw} \leftarrow \max\left(\min\left(q_w, L_w - 1 - \frac{W_w}{2}\right), \frac{W_w}{2}\right)$

**Compute masks:**

time_constraint $\leftarrow |q_{ct} - k_t| \leq \frac{W_t}{2}$

hori_constraint $\leftarrow |q_{ch} - k_h| \leq \frac{W_h}{2}$

vert_constraint $\leftarrow |q_{cw} - k_w| \leq \frac{W_w}{2}$

**return** time_constraint $\wedge$ hori_constraint $\wedge$ vert_constraint

---

**Mask of 3D STA** Algorithm 3 defines the mask for STA, introducing a tile-based coordinate framework that differs from 3D NATTEN. First, query and key coordinates are mapped to tile coordinates, where each QK pair is assigned a tile ID, with queries and keys in the same tile sharing the same ID. STA also computes the window center within tile coordinates, ensuring queries remain within valid bounds. Finally, neighboring keys are selected based on their tile distance from the query's window center.

**Tiling in STA** Figure 8 illustrates STA's token tiling and ordering mechanism in a 2D scenario, which extends naturally to 3D. Unlike conventional approaches that flatten 2D/3D data into 1D sequences using a zigzag pattern, STA organizes tokens into tiles, ensuring that tokens within a tile maintain neighboring sequence IDs. This ordering strategy preserves locality, so when a tile attends to another tile, the resulting attention map forms a dense block, as all participating sequence IDs remain consecutive.

**Visualization of 2D SWA** In Figure 9, we illustrate how query tokens attend to its window key tokens. In 2D-SWA, the window slides token by token. For each window, SWA calculates the attention between the center q with all keys within the window.

---

**Algorithm 3** Mask Definition of 3D STA

---

**Input:** Query coordinates $(q_t, q_h, q_w)$, key coordinates $(k_t, k_h, k_w)$, video size $(L_t, L_h, L_w)$, kernel size $(W_t, W_h, W_w)$, tile size $(T_t, T_h, T_w)$

**Compute QK coordinates in:**

$q_{t,\text{tile}} \leftarrow q_t // T_t$

$q_{h,\text{tile}} \leftarrow q_h // T_h$

$q_{w,\text{tile}} \leftarrow q_w // T_w$

$k_{t,\text{tile}} \leftarrow k_t // T_t$

$k_{h,\text{tile}} \leftarrow k_h // T_h$

$k_{w,\text{tile}} \leftarrow k_w // T_w$

**Compute window size in tiles:**

$W_{t,\text{tile}} \leftarrow W_t // T_t$

$W_{h,\text{tile}} \leftarrow W_h // T_h$

$W_{w,\text{tile}} \leftarrow W_w // T_w$

**Compute window center:**

$q_{ct} \leftarrow \max\left(\min\left(q_{t,\text{tile}}, (L_t // T_t - 1) - \frac{W_{t,\text{tile}}}{2}\right), \frac{W_{t,\text{tile}}}{2}\right)$

$q_{ch} \leftarrow \max\left(\min\left(q_{h,\text{tile}}, (L_h // T_h - 1) - \frac{W_{h,\text{tile}}}{2}\right), \frac{W_{h,\text{tile}}}{2}\right)$

$q_{cw} \leftarrow \max\left(\min\left(q_{w,\text{tile}}, (L_w // T_w - 1) - \frac{W_{w,\text{tile}}}{2}\right), \frac{W_{w,\text{tile}}}{2}\right)$

**Compute masks:**

time_constraint $\leftarrow |q_{ct} - k_{t,\text{tile}}| \leq \frac{W_{t,\text{tile}}}{2}$

hori_constraint $\leftarrow |q_{ch} - k_{h,\text{tile}}| \leq \frac{W_{h,\text{tile}}}{2}$

vert_constraint $\leftarrow |q_{cw} - k_{w,\text{tile}}| \leq \frac{W_{w,\text{tile}}}{2}$

**return** time_constraint $\wedge$ hori_constraint $\wedge$ vert_constraint

---

## B. Finetuning Details

We train on 2,000 synthetically generated videos from HunyuanVideo at a resolution of 1280×768 with 117 frames. The prompts are sourced from the Mixkit dataset (Lin et al., 2024). To reduce memory usage and accelerate training, we precompute VAE-encoded latents and text encoder states. Training involves fine-tuning for 1,600 steps with a batch size of 2 and a learning rate of 2e-5. We optimize using the loss function from Eq. (5) with coefficients $\alpha = 1$, $\beta = 0.5$, and $\gamma = 0.5$. To prevent overfitting on a single guidance scale, we alternate between guidance scales of 1 and 6 at odd and even steps. The entire process runs on 8 H100 GPUs with FSDP and context parallelism for training (8 hours) and sequence parallelism for inference.

## C. Further Details of Baselines

**Swin Transformer** (Liu et al., 2021b) introduces a hierarchical vision transformer with a shifted window-based attention mechanism. Instead of computing self-attention globally, it partitions the image into non-overlapping windows and applies attention locally, improving computational efficiency. A key innovation is the alternating window partitioning strategy: one layer uses standard window partitioning, while the next shifts the windows to enable cross-window connections

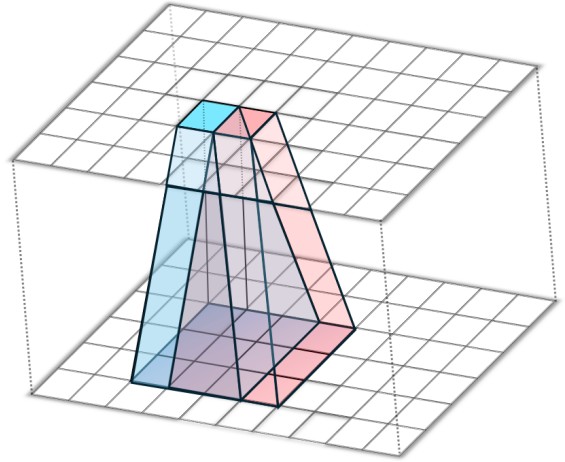

*Figure 8. Left*: Conventional zigzag flattening strategy. *Right*: STA' sequence flattening strategy. The plot is given assuming a (9, 9) image with (3, 3) tile size.

*Figure 9.* 2D Sliding Window Attention visualization.

and better information exchange. Swin attention is typically used in a train-from-scratch setting. A limitation of this approach is that it disrupts local connectivity within a single attention layer. Tokens in adjacent regions may not attend to each other if they fall into separate windows. In this paper, we apply Swin attention to HunyuanVideo and shift the window every other layer accordingly.

**CLEAR** (Liu et al., 2024) achieves linear attention by replacing the original full attention with a circular window-based attention mechanism where each query token only attends to key-value tokens within a radius r, maintaining the same scaled dot-product attention formula but restricting its computation to local windows. The authors implement CLEAR with FlexAttention.

$\Delta$**-DiT** (Chen et al., 2024) optimizes inference speed by caching feature offsets instead of full feature maps. It employs a staged caching strategy: residuals from later DiT blocks are stored for early-step sampling, while residuals from earlier blocks are cached for later steps. The key parameters in $\Delta$-DiT include the residual cache interval $N$, the number of cached blocks $N_c$, and the timestep boundary $b$, which determines the cache position. Since the official

*Table 5.* Training-free performance on Wan 2.1

| Model | SSIM ↑ | PSNR ↑ | Latency | Speedup |
|-------|--------|--------|---------|---------|
| STA | 85.81 | 24.42 | 730s | 1.60× |

$\Delta$-DiT implementation is unavailable, we reimplemented its method based on the paper to accelerate video generation. Given a speedup budget, we vary $N_c$, $N$, and $b$ to pick the best hyperparameters, ensuring a fair evaluation of its effectiveness. For the 50-step 1.36× speedup, we set $N_c = 24$, $N = 3$, and $b = 24$. For 1.8× speedup, we set $N_c = 28$, $N = 6$, and $b = 24$.

## D. Results on Wan 2.1

Following Table 3, we evaluate the effectiveness of STA on Wan 2.1. Despite being applied to a different model, STA achieves comparable performance across key evaluation metrics while preserving the same sparsity level. Wan 2.1 is evaluated on videos with the same resolution but a shorter length of 69 frames, resulting in a reduced end-to-end speedup.

## E. Results on Image Super-Resolution

*Table 6.* Image superresolution results with FLUX (Black-Forest, 2023) on 1000 captions randomly sampled from COCO-2014 (Lin et al., 2015) validation dataset.

| Methods | SSIM | PSNR | Sparsity | Latency | Speedup |
|---------|------|------|----------|---------|---------|
| *1K→2K* | | | | | |
| CLEAR r=16 | 0.9291 | 28.1142 | 96.12% | 13s | 1.54× |
| CLEAR r=32 | 0.9443 | 29.6722 | 85.94% | 15s | 1.33× |
| STA w=(48,72) | 0.9357 | 29.1086 | 81.25% | 14s | 1.43× |
| *2K→4K* | | | | | |
| CLEAR r=16 | 0.9394 | 29.0463 | 98.98% | 67s | 2.90× |
| CLEAR r=32 | 0.9455 | 30.0742 | 96.08% | 92s | 2.11× |
| STA w=(48,72) | 0.9470 | 30.1939 | 95.31% | 57s | 3.40× |

## F. More Experiment Results

### F.1. Kernel Performance

We additionally benchmark various sparse attention kernels at a sparsity level of around 56% and present the results in Table 7. With lower sparsity, sparse kernels generally have a higher MFU, but the findings in Table 2 remain unchanged.

### F.2. Detailed VBench Results

In Tables 8 and 9, we present detailed comparisons of VBench scores across key dimensions originally summarized in Table 4. Our analysis reveals that STA surpasses swin attention in video quality metrics such as Imaging Qual-

*Table 7.* Speedup with sparse attention kernels on H100.

| Methods | Implementation | Config | Sparsity | TFLOPS | Latency(ms) | MFU | Kernel Efficiency | Speedup |
|---|---|---|---|---|---|---|---|---|
| FA 3 | ThunderKittens | - | 0.00% | 164.03 | 265.28 | 62.49% | 100.00% | 1.00× |
| FA 3 | CUDA | - | 0.00% | 164.03 | 256.59 | 64.61% | 103.39% | 1.03× |
| CLEAR | FlexAttention | r=32 | 56.23% | 71.80 | 675.05 | 10.75% | 17.20% | 0.39× |
| NATTEN | FlexAttention | w=(30,41,41) | 56.22% | 71.81 | 804.62 | 9.02% | 14.43% | 0.33× |
| Tiled NATTEN | CUDA | w=(29,41,41) | 57.68% | 69.41 | 173.57 | 4.04% | 6.47% | 0.15x |
| Tiled NATTEN | FlexAttention | w=(30,41,41) | 56.22% | 71.81 | 409.89 | 17.70% | 28.33% | 0.65× |
| Swin | FlexAttention | w=(48,64,64) | 55.81% | 72.49 | 127.51 | 57.46% | 91.95% | 2.08× |
| STA | FlexAttention | w=(30,40,40) | 58.33% | 68.35 | 174.17 | 39.66% | 63.46% | 1.52× |
| STA | ThunderKittens | w=(30,40,40) | 58.33% | 68.35 | 111.73 | 61.82% | 98.93% | 2.37× |

ity and Multiple Objects, while achieving comparable or superior scores to CLEAR and Tiled NATTEN. For training-free models, we observe a systematic degradation in quality-related metrics (e.g., temporal flickering, motion smoothness) as sparsity increases in the STA attention mechanism. Conversely, semantic-aligned dimensions—including Appearance Style, Color, and Spatial Relationships—improve under higher sparsity regimes, a phenomenon driven by the text embeddings' amplified role in attention computation when spatial-temporal attention is sparsified. Furthermore, the trained STA demonstrates significant gains in video quality metrics over its untrained counterpart, while maintaining semantic coherence at comparable levels which underscores the efficacy of training in refining low-level visual fidelity without compromising text-video alignment.

## G. Qualitative Examples

We show qualitatively show videos generated by the original HunyuanVideo, STA, and $\Delta$-DiT in Figure 10 and Figure 11. While fine-tuning introduces minor shifts in the output distribution of STA-t-2.43x, the model still preserves high video generation quality. Videos generated by $\Delta$-DiT are generally less sharp than those generated by the original HunyuanVideo and STA. More demos are available at https://fast-video.github.io/.

*Table 8.* Model Performance Comparison - Part 1

| Model | Appearance Style | Subject Consistency | Background Consistency | Temporal Flickering | Motion Smoothness | Dynamic Degree | Aesthetic Quality | Imaging Quality | Overall Consistency |
|---|---|---|---|---|---|---|---|---|---|
| FA3 | 18.43% | 94.22% | 96.74% | 99.21% | 99.15% | 75.00% | 64.63% | 67.97% | 25.96% |
| *w.o training* | | | | | | | | | |
| CLEAR | 18.73% | 93.63% | 96.51% | 98.99% | 99.01% | 68.06% | 63.75% | 68.35% | 26.23% |
| Tiled NATTEN | 18.79% | 94.59% | 96.61% | 98.75% | 98.85% | 70.83% | 63.79% | 68.16% | 26.53% |
| Swin w=(48,64,64) | 20.85% | 91.74% | 95.48% | 98.67% | 97.77% | 77.78% | 51.01% | 62.22% | 25.27% |
| Swin w=(30,40,40) | 20.62% | 90.33% | 93.09% | 98.78% | 96.53% | 75.00% | 48.10% | 61.89% | 25.62% |
| STA w=(30,40,40) | 18.79% | 94.75% | 96.50% | 98.82% | 98.83% | 69.44% | 64.18% | 68.39% | 26.47% |
| STA w=(18,24,24) | 21.25% | 89.66% | 91.64% | 98.46% | 97.27% | 83.33% | 59.75% | 64.23% | 26.61% |
| *w. training* | | | | | | | | | |
| Swin w=(30,40,40) | 20.07% | 89.78% | 94.93% | 98.86% | 96.64% | 70.83% | 44.91% | 55.99% | 26.00% |
| STA w=(30,24,40) | 18.90% | 94.90% | 97.60% | 99.68% | 99.23% | 73.61% | 63.77% | 66.21% | 26.58% |
| STA w=(18,24,24) | 18.90% | 94.64% | 96.76% | 99.22% | 99.11% | 69.44% | 64.52% | 66.67% | 26.09% |

*Table 9.* Model Performance Comparison - Part 2

| Model | Object Classification | Multiple Objects | Human Action | Color | Spatial Relationship | Scene | Quality Score | Semantic Score | Final Score |
|---|---|---|---|---|---|---|---|---|---|
| FA3 | 85.76% | 70.12% | 90.00% | 88.66% | 71.28% | 35.25% | 85.34% | 72.17% | 82.71% |
| *w.o training* | | | | | | | | | |
| CLEAR | 88.13% | 77.97% | 88.00% | 91.10% | 77.49% | 32.85% | 84.41% | 74.20% | 82.37% |
| Tiled NATTEN | 83.54% | 72.18% | 94.00% | 92.28% | 81.21% | 37.94% | 84.61% | 75.00% | 82.69% |
| Swin w=(48,64,64) | 78.16% | 58.54% | 87.00% | 93.68% | 77.45% | 37.79% | 80.91% | 71.35% | 79.00% |
| Swin w=(30,40,40) | 79.19% | 60.44% | 88.00% | 93.68% | 77.24% | 35.54% | 78.84% | 72.28% | 77.53% |
| STA w=(30,40,40) | 80.54% | 71.19% | 93.00% | 89.81% | 79.25% | 36.77% | 84.63% | 73.83% | 82.47% |
| STA w=(18,24,24) | 88.13% | 75.46% | 91.00% | 91.61% | 82.52% | 42.15% | 81.47% | 77.03% | 80.58% |
| *w. training* | | | | | | | | | |
| Swin w=(30,40,40) | 77.14% | 48.86% | 73.00% | 87.00% | 63.38% | 39.03% | 77.50% | 67.39% | 75.48% |
| STA w=(30,24,40) | 91.77% | 68.45% | 86.00% | 89.59% | 72.76% | 39.53% | 85.37% | 73.52% | 83.00% |
| STA w=(18,24,24) | 92.96% | 74.16% | 93.00% | 84.50% | 73.41% | 38.23% | 84.76% | 74.05% | 82.62% |

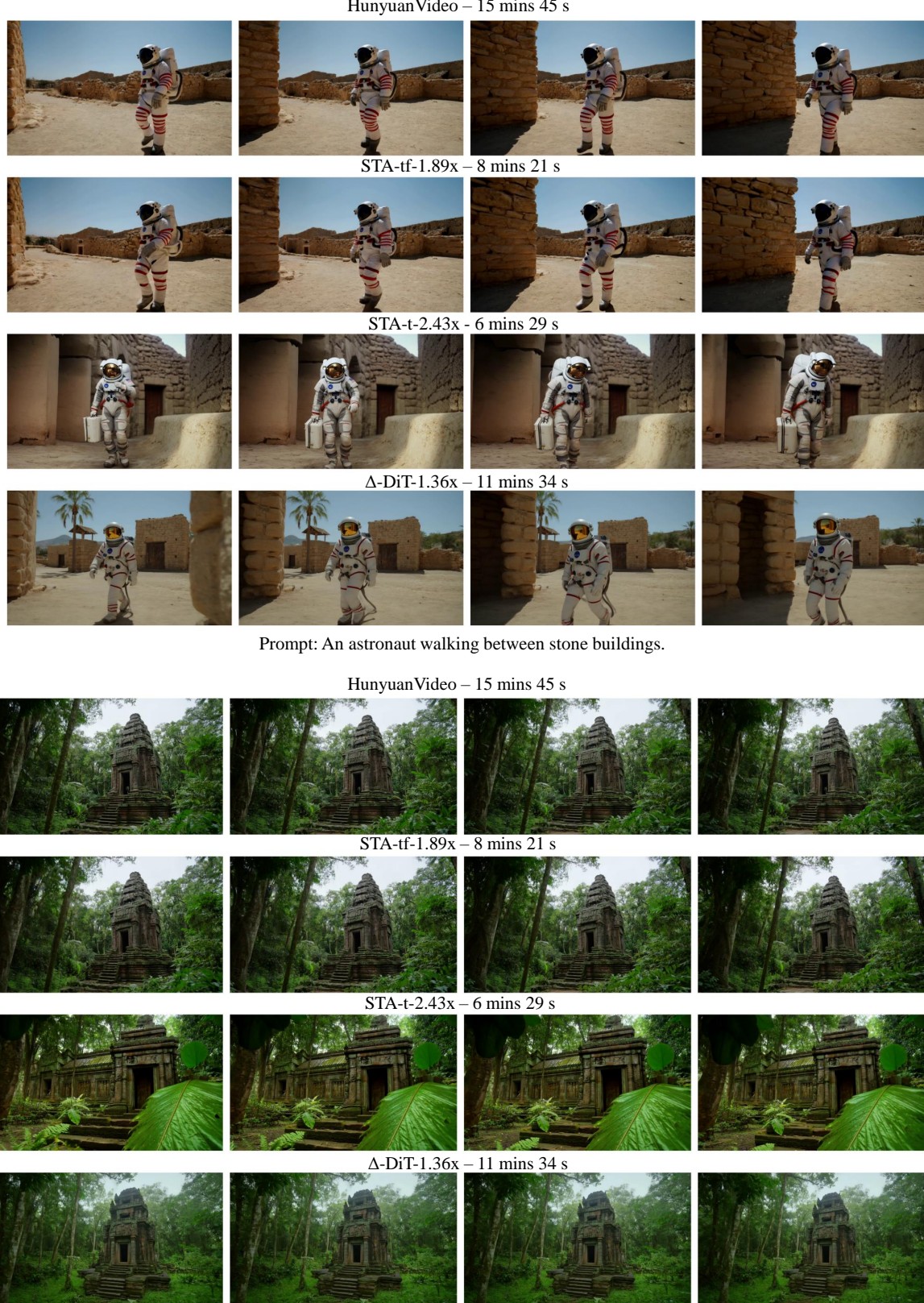

HunyuanVideo – 15 mins 45 s

STA-tf-1.89x – 8 mins 21 s

STA-t-2.43x – 6 mins 29 s

Δ-DiT-1.36x – 11 mins 34 s

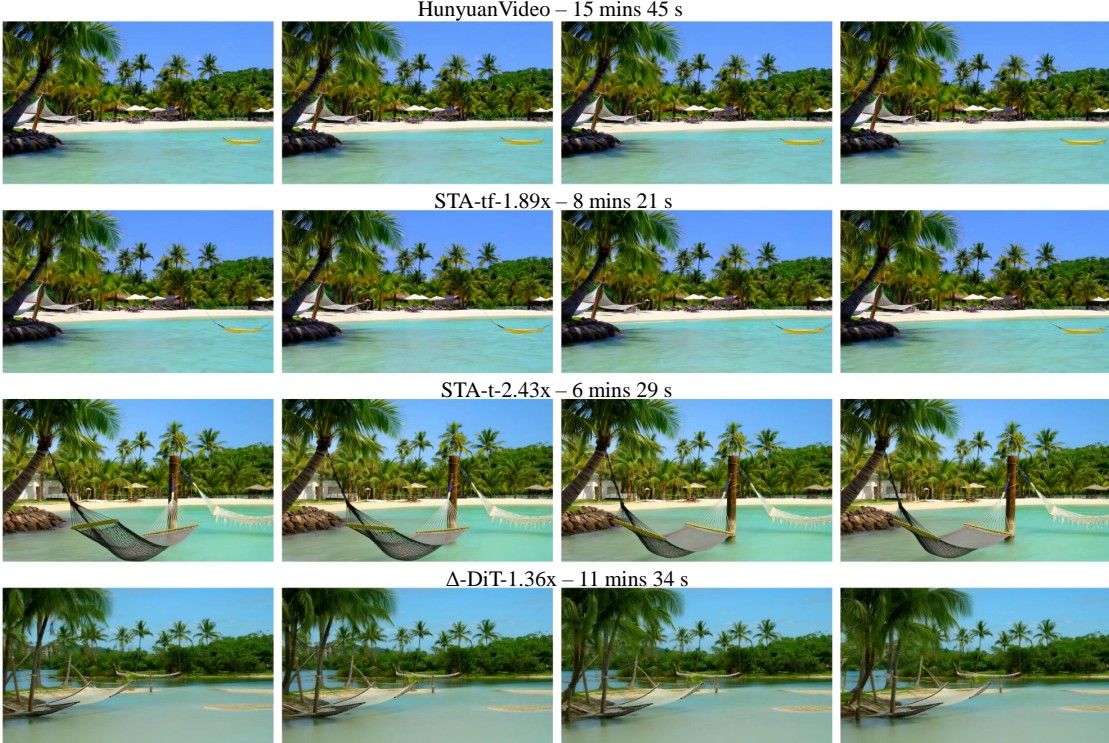

Prompt: A tranquil island retreat features swaying palm trees and hammocks strung between them, inviting guests to relax and enjoy the serene beauty of the surroundings.

HunyuanVideo – 15 mins 45 s

STA-tf-1.89x – 8 mins 21 s

STA-t-2.43x – 6 mins 29 s

Δ-DiT-1.36x – 11 mins 34 s

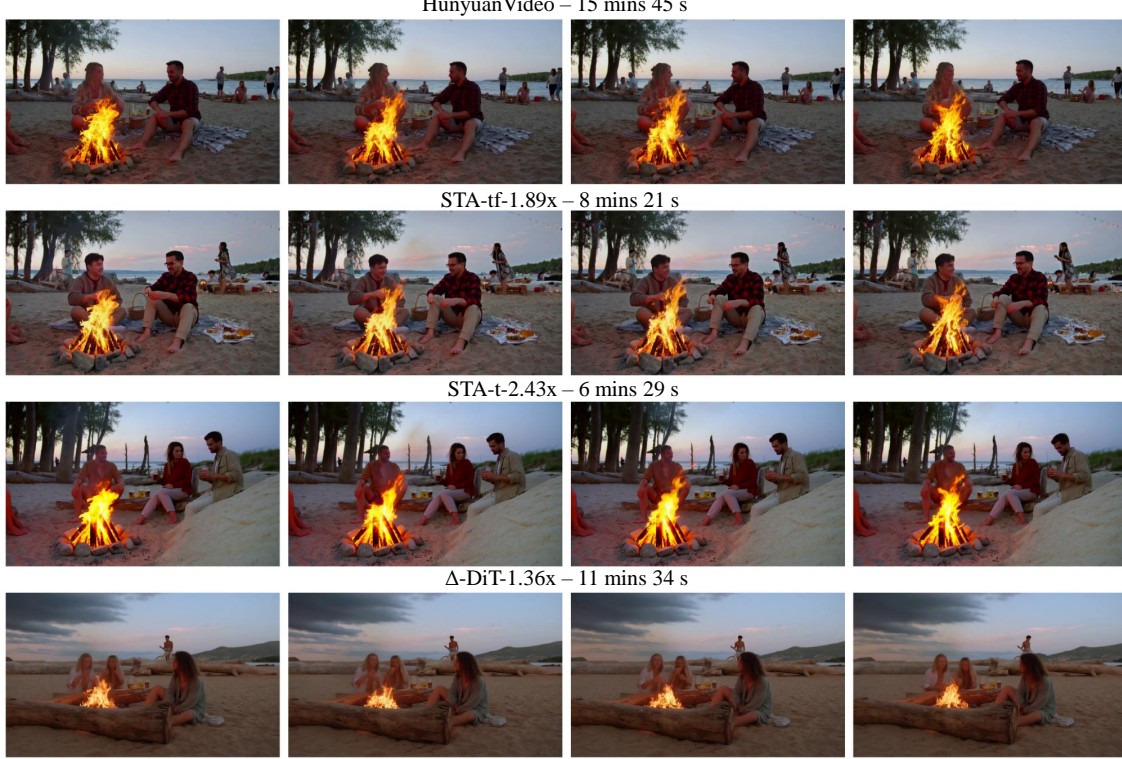

Prompt: People gather on a peaceful beach at sunset, a bonfire crackling as they sit around, enjoying the warmth and the sight of the sun dipping below the horizon.

*Figure 11.* Qualitative comparisons. While fine-tuning introduces minor shifts in the output distribution of STA-t-2.43x, the model still preserves high video generation quality. Videos generated by Δ-DiT are generally less sharp than those generated by the original HunyuanVideo and STA.

