# OpenReview forum: "Fast Video Generation with Sliding Tile Attention"
_ICML.cc/2025/Conference — ICML 2025 poster_

### Official Review · Reviewer_BHCc · 2025-03-06

**Overall Recommendation:** 4

**Summary:**

this paper addresses the problem of slow speeds in video generation. the paper proposes a method called sliding tile attention that is designed to address the challenge. the proposed method learns to do sliding and attending over local spatial and temporal region, allowing  the  reduction of redundancy in computing full attention. the resulting method is much faster than prior methods.

**Claims And Evidence:**

the claims are supported by experimental evaluations.

**Essential References Not Discussed:**

references are adequate

**Experimental Designs Or Analyses:**

the evaluation is thorough.

**Methods And Evaluation Criteria:**

the proposed method and evaluation metrics make sense

**Other Comments Or Suggestions:**

n/a

**Other Strengths And Weaknesses:**

i enjoyed reading this paper and i believe this will be useful for many applications. however, I would still like to request some clarifications

1. will the code be released to ensure reproducibility

2. will the proposed method have the potential to be applied to 3D or 4D generation? (I know 3D and 4D gen is beyond the scope of this paper, so I'm not asking for any experiments or comparisons, but I'm genuinely curious about the impact that this method could potentially have

**Questions For Authors:**

please see the comments in #Other Strengths And Weaknesses

**Relation To Broader Scientific Literature:**

i believe the resulting method will have a huge impact on many researchers working on diffusion models. i see the merit of the proposed method and the potential impact this paper could have

**Theoretical Claims:**

the theorems are technically sound

---

> ### Author Rebuttal · Authors · 2025-03-31
>
> We sincerely thank Reviewer BHCc for their supportive feedback and genuine interest in our work.
>
> > *Will the code be released to ensure reproducibility*
>
> We appreciate your emphasis on reproducibility. We confirm that all the code, configurations, and scripts required to reproduce our experiments will be fully open-sourced. This will ensure the community can easily replicate and build upon our results.
>
> > *Will the proposed method have the potential to be applied to 3D or 4D generation?*
>
> We appreciate your insightful question regarding STA's potential applicability beyond standard video generation. Videos are already a type of 3D data (time × height × width). We believe STA naturally generalizes to other forms of 3D data such as 3D object generation (e.g., voxel grids) or even 4D data generation (e.g., dynamic point clouds or time-evolving 3D structures), so long as those data conform to some degree of data locality.

---

### Official Review · Reviewer_nmHo · 2025-03-11

**Overall Recommendation:** 3

**Summary:**

This paper introduces sliding tile attention (STA) to address prohibitive compute cost in attention calculation. The authors observed that attention scores in pretrained video diffusion models predominantly concentrate within localized 3D windows. The proposed STA can eliminate redundancy from full attention by sliding and attending over local spatial-temporal region.  STA operates tile-by-tile with a novel hardware-aware sliding window design, preserving expressiveness while being hardware-efficient. Experimental results verify the effectiveness of the proposed method in achieving diffusion acceleration.

**Claims And Evidence:**

This paper claims that "attention scores in pretrained video diffusion models predominantly concentrate within localized 3D windows.". The claims are supported by experimental results.

**Essential References Not Discussed:**

No

**Experimental Designs Or Analyses:**

Yes. STA operates tile-by-tile with a novel hardware-aware sliding window design, preserving expressiveness while being hardware-efficient.

**Methods And Evaluation Criteria:**

This paper introduces sliding tile attention (STA) to address prohibitive compute cost in attention calculation. Both qualitative and quantitative results verify the effectiveness of the proposed method.

**Other Comments Or Suggestions:**

Please refer to Weaknesses for more details

**Other Strengths And Weaknesses:**

Pros:

1.	The authors observed that attention scores in pretrained video diffusion models predominantly concentrate within localized 3D windows.
2.	A novel hardware-aware sliding window design is proposed , preserving expressiveness while being hardware-efficient
3.	Extensive experimental results verify the effectiveness of the proposed method.

Cons:

1.	More training details about STA w. Training are missing.

2.	No demo videos are provided in the supplementary files, which makes it hard to judge the effectiveness of the proposed method.

3.	For video generation, only HunyuanVideo is used to conduct experiments. More other architectures such as CogVideoX and Mochi should be discussed and compored.

**Questions For Authors:**

Please refer to Weaknesses for more details

**Relation To Broader Scientific Literature:**

This paper introduces sliding tile attention (STA) to address prohibitive compute cost in attention calculation. The authors observed that attention scores in pretrained video diffusion models predominantly concentrate within localized 3D windows. The proposed STA can eliminate redundancy from full attention by sliding and attending over local spatial-temporal region.  STA operates tile-by-tile with a novel hardware-aware sliding window design, preserving expressiveness while being hardware-efficient. Experimental results verify the effectiveness of the proposed method in achieving diffusion acceleration.

**Theoretical Claims:**

Yes

---

> ### Author Rebuttal · Authors · 2025-03-31
>
> We thank the reviewer nmHo for their constructive feedback. Below, we address specific comments one by one.
>
> > *More training details about STA w. Training are missing.*
>
> We have provided the primary details of STA training, including datasets, prompts, learning rate, and hardware specifications, in Appendix Section B and briefly mentioned them in line 265 of the manuscript. If there are specific additional details you would like us to include beyond these, we would be happy to provide them. We will also open-source all of our **training code, configurations, and scripts** to facilitate reproducibility.
>
> > *No demo videos are provided in the supplementary files*
>
> Thanks for raise this concern. We provide an anonymous link to 20 sampled videos from our human evaluation experiments here: https://drive.google.com/drive/u/4/folders/1kRDt4ahiYQj1zk593FE6Hg2CgNsJcdIf
>
> > *For video generation, only HunyuanVideo is used to conduct experiments.*
>
> Following your valuable suggestion, we further validated STA on Wan2.1 in a training-free setup, as shown in the results below. Together with FLUX's result in Table 5 of our manuscript, we believe they provide broader evidence of STA's generalizability.
>
>
> Resolution: 93X1024X1024
>
>
> | Steps  |  Sparsity                  | SSIM ↑ | PSNR ↑ | Latency | Speedup |
> |------------|-----------------------------------|--------|--------|---------|---------|
> | 50 steps full attn|   0.0%    |  –|   –   | 1996s    | 1.00×   |
> | 18 steps full + 32 steps STA|   50.31%    | 77.41 | 20.16  | 1322s    | 1.51×   |
> | 25 steps full attn|   0.0%    |  –|   –   | 998s    | 1.00×   |
>  | 9 steps full + 16 steps STAl |  49.83%    | 79.3% | 22.29  | 661s    | 1.51×   |
>  | 10  steps full attn|   0.0%    |  –|   –   | 402s    | 1.00×   |
> | 4 steps full + 6 steps STA |  46.61%    | 82.7% | 22.25  | 277s    | 1.45×   |
>
>
>
>
> Resolution: 93X768X1280
>
>
>
>
> | Steps                    | Sparsity     | SSIM ↑ | PSNR ↑ | Latency | Speedup |
> |--------------------------|--------------|--------|--------|---------|---------|
> | 50 steps full attn       | 0.0%         | –      | –      | 1839s   | 1.00×   |
> | 18 steps full + 32 STA   | 52.71%       | 80.97 | 22.09  | 1241s   | 1.48×   |
> | 25 steps full attn       | 0.0%         | –      | –      | 920s    | 1.00×   |
> | 9 steps full + 16 STA    | 52.36%       | 79.48  | 22.3   | 621s      | 1.48×   |
> | 10 steps full attn       | 0.0%         | –      | –      | 378s    | 1.00×   |
> | 4 steps full + 6 STA     | 49.13%       | 79.49  | 22.73  | 258s    | 1.46×   |
>
>
> The results confirm STA's applicability across different model architectures.
>
> We truly appreciate your thoughtful review. If our response has addressed your concerns, we would be grateful if you might consider a higher score. If there are any remaining points we could further clarify or improve, we would be sincerely thankful for your guidance.

---

### Official Review · Reviewer_C6tW · 2025-03-12

**Overall Recommendation:** 4

**Summary:**

This paper introduces sliding tile attention (STA) that operates tile-by-tile with a novel hardware-aware sliding window design, preserving expressiveness while being hardware-efficient. STA achieves 1.36–3.53× end-to-end speedup with no or minimum quality loss.

## update after rebuttal"
Thanks for the authors' response. The authors have addressed most of my concerns. My final score is 4 "accept".

**Claims And Evidence:**

Yes, claims made in the submission are supported by clear and convincing evidence.

**Essential References Not Discussed:**

I don't see any specific missing pieces but it would be helpful to extend the current relative work section with more literatures on attention based speed-up that are more related to the paper, instead of other types of speed-up.

**Experimental Designs Or Analyses:**

Yes, the experiments overall are sound

**Methods And Evaluation Criteria:**

Yes, proposed methods and evaluation criteria make sense for the problem

**Other Comments Or Suggestions:**

Extend the current relative work section with more literatures on attention based speed-up that are more related to the paper, instead of other types of speed-up.

It would be great to present results on more models / hardware types to show case the generalizability of the work.

**Other Strengths And Weaknesses:**

The paper overall is well written and easy to follow. Experiments are sound, contribution of the paper is clear. The topic is very important to the community. Overall it's a good paper.

**Questions For Authors:**

Are the results generalizable to other models than Huanyuan-Video? E.g., StepVideo, Wan, CogVideoX, LTX. Can you please provide some analysis on those models?

Are the optimization hardware dependent? It would be good to report speed-ups on other types of GPUs.

**Relation To Broader Scientific Literature:**

Video generation model is running with very high latency. Reducing latency is a very significant topic to the area. This paper introduces some innovations along the direction.

**Theoretical Claims:**

Yes, overall looks correct

---

> ### Author Rebuttal · Authors · 2025-03-31
>
> We sincerely thank Reviewer C6tW for their insightful suggestions and valuable questions. Below we address your comments and strengthen our paper.
>
> > *Are the results generalizable to other models than Huanyuan-Video?*
>
>
>
>
> As suggested, we further validated STA's generalizability beyond Huanyuan-Video. In our original manuscript, we demonstrated the effectiveness of STA on image-generation models like FLUX. Here, we additionally apply STA to Wan 2.1 in a training-free setup.
>
>
> Resolution: 93X1024X1024
>
>
> | Steps  |  Sparsity                  | SSIM ↑ | PSNR ↑ | Latency | Speedup |
> |------------|-----------------------------------|--------|--------|---------|---------|
> | 50 steps full attn|   0.0%    |  –|   –   | 1996s    | 1.00×   |
> | 18 steps full + 32 steps STA|   50.31%    | 77.41 | 20.16  | 1322s    | 1.51×   |
> | 25 steps full attn|   0.0%    |  –|   –   | 998s    | 1.00×   |
>  | 9 steps full + 16 steps STAl |  49.83%    | 79.3% | 22.29  | 661s    | 1.51×   |
>  | 10  steps full attn|   0.0%    |  –|   –   | 402s    | 1.00×   |
> | 4 steps full + 6 steps STA |  46.61%    | 82.7% | 22.25  | 277s    | 1.45×   |
>
>
>
>
> Resolution: 93X768X1280
>
>
>
>
> | Steps                    | Sparsity     | SSIM ↑ | PSNR ↑ | Latency | Speedup |
> |--------------------------|--------------|--------|--------|---------|---------|
> | 50 steps full attn       | 0.0%         | –      | –      | 1839s   | 1.00×   |
> | 18 steps full + 32 STA   | 52.71%       | 80.97 | 22.09  | 1241s   | 1.48×   |
> | 25 steps full attn       | 0.0%         | –      | –      | 920s    | 1.00×   |
> | 9 steps full + 16 STA    | 52.36%       | 79.48  | 22.3   | 621s      | 1.48×   |
> | 10 steps full attn       | 0.0%         | –      | –      | 378s    | 1.00×   |
> | 4 steps full + 6 STA     | 49.13%       | 79.49  | 22.73  | 258s    | 1.46×   |
>
>
> The above results confirm the generalizability of STA beyond Huanyuan-Video. On Wan 2.1, STA achieves consistent speed-ups across different step settings and resolutions without compromising perceptual quality.
>
>
>
> > *Are the optimization hardware dependent?*
>
> STA itself is not hardware dependent, and we designed our method to be easily deployable across GPU architectures:
>
>
> **A100/RTX 4090**
>
>
> Using the FlexAttention backend, STA achieves substantial speedups with minimal overhead, as demonstrated in the table below:
>
>
> RTX 4090:
>
>
> | Methods   | Implementation | Config        | Sparsity | TFLOPS | Latency(ms) | MFU    | Kernel Efficiency | Speedup |
> |-----------|----------------|---------------|----------|--------|-------------|--------|-------------------|---------|
> | Full Attn | FA2            | -             | 0.00%    | 164.03 | 958.68      | 51.85% | 100.00%           | 1.00×   |
> | Full Attn | Flex Attn      | -             | 0.00%    | 164.03 | 984.16      | 50.51% | 97.41%            | 0.97×   |
> | STA       | Flex Attn      | wt=(18,24,24) | 91.00%   | 14.76  | 896.74      | 49.89% | 96.22%            | 10.69×  |
>
>
>
>
> A100:
> | Methods | Implementation | Config | Sparsity | TFLOPS | Latency(ms) | MFU | Kernel Efficiency | Speedup |
> |-----------|----------------|---------------|----------|--------|-------------|--------|-------------------|---------|
> | Full Attn | FA2 | - | 0.00% | 164.03 | 697.61 | 75.36% | 100.00% | 1.00× |
> | Full Attn | Flex Attn | - | 0.00% | 164.03 | 999.03 |52.63% | 69.83% | 0.70× |
> | STA | Flex Attn | wt=(18,24,24) | 91.00% | 14.76 | 89.97 | 52.59% | 69.78% | 7.75× |
>
>
>
>
> **On H100 GPUs**
>
>
> We further leverage Tensor Memory Accelerator (TMA) capabilities to achieve even better performance. Specifically, STA implemented in TK on H100 achieves a 1.43× improvement compared to FlexAttention implementation, showcasing additional optimization potential with modern hardware.
>
>
> In summary, STA does not inherently depend on specific GPU architectures. While advanced hardware features like TMA can further boost STA's performance, using FlexAttention alone provides straightforward and efficient speedups across other GPUs.
>
> > *Extend the current relative work section with more literatures on attention based speed-up that are more related to the paper, instead of other types of speed-up.*
>
> We appreciate your suggestion to enhance the related work section of our manuscript. Should the paper be accepted and additional space permit, we plan to incorporate discussions on the following areas:​ 1. Efficient Vision Transformers (ViTs) with a focus on attention mechanisms such as Swin Transformer and EfficientViT. 2. Efficient attention mechanisms in large language models such as  StreamingLLM and H2O.​ 3. Attention quantization techniques such as SageAttention.​

---

### Official Review · Reviewer_3sP9 · 2025-03-14

**Overall Recommendation:** 3

**Summary:**

This paper introduces Sliding Tile Attention (STA), a novel attention mechanism designed to accelerate video generation using Diffusion Transformers (DiTs). The key idea is to leverage the observation that attention scores in pretrained video diffusion models are predominantly concentrated within localized 3D windows, thus eliminating redundancy from full attention. STA operates tile-by-tile with a hardware-aware sliding window design, preserving expressiveness while being hardware-efficient. The paper claims that STA achieves significant speedups (1.36–3.53× end-to-end) with minimal or no quality loss compared to existing methods like FlashAttention-3.

**Claims And Evidence:**

The claims made in the submission are supported by clear and convincing evidence. The authors provide extensive experimental results, including efficiency metrics (MFU and latency), human evaluations, and automated metrics (VBench, SSIM, PSNR, CD-FVD). The evidence demonstrates that STA achieves significant speedups with minimal quality loss. The paper also includes detailed comparisons with baseline methods like CLEAR, NATTEN, and Swin, showing that STA outperforms these methods in both efficiency and quality.

**Essential References Not Discussed:**

None

**Experimental Designs Or Analyses:**

The experimental designs and analyses are sound and valid. The authors conducted extensive experiments to evaluate the efficiency and quality of STA. They benchmarked the efficiency of various attention algorithms, including STA, against baseline methods using metrics like MFU and latency. Human evaluations were performed on a large set of prompts to assess the quality of generated videos.

**Methods And Evaluation Criteria:**

The proposed methods and evaluation criteria make sense for the problem of accelerating video generation with DiTs. The STA mechanism is designed to address the computational overhead of 3D full attention by leveraging localized attention patterns. The evaluation criteria include both efficiency metrics  and quality metrics.The methods are well-motivated and the evaluation is comprehensive.

**Other Comments Or Suggestions:**

None

**Other Strengths And Weaknesses:**

None

**Questions For Authors:**

I think STA should have restrictions on input resolution, length, and aspect ratio. The authors should clarify this key issue.

**Relation To Broader Scientific Literature:**

NAN

**Theoretical Claims:**

The theoretical claims in the paper are correct. The authors provide a clear formulation of the STA mechanism, including the tiling strategy and the attention mask definition. The theorems (3.1 and 3.2) are used to quantify the number of dense and mixed blocks in different attention configurations, which helps in understanding the efficiency gains of STA.

---

> ### Author Rebuttal · Authors · 2025-03-31
>
> We appreciate Reviewer 3sP9's insightful question regarding the constraints on input resolutions and aspect ratios for Sliding Tile Attention (STA). Below, we clarify and expand upon these points:
>
> > *I think STA should have restrictions on input resolution, length, and aspect ratio. The authors should clarify this key issue.*
>
> Indeed, STA requires input video latents to have dimensions that are integer multiples of the tile size to fully achieve the promised efficiency gains from sparsity. We briefly mentioned this constraint in line 214 of our submission. Specifically, in our current implementation, we use a tile size of (6, 8, 8), meaning the optimal latent dimensions are multiples of these numbers, i.e., (a × 6, b × 8, c × 8) where a, b, and c are positive integers.
>
>
> For video latents whose dimensions do not strictly satisfy this requirement, we suggest two practical approaches to address this limitation:
>
> **Padding with Masking**:
>
> When generating a video with a dimension slightly different from multiples of the tile size, we can introduce padding tokens to match the tile-aligned dimension (which is a standard practice that is also used in FlashAttention kernel when the input sequence length is not an integer multiple of block size.) For instance, to generate a video with dimensions (29, 45, 76), we would pad the latents to size (30, 48, 80).  In practice, padding tokens are then masked during attention computation to avoid contaminating the attention.
>
> **Cropping**:
>
> Alternatively, one can generate a slightly larger latent (e.g.,(30, 48, 80)) and subsequently crop the result to the desired dimensions.
>
>
> Although both methods introduce computational overhead due to processing padding tokens or larger dimensions, this cost remains minimal compared to STA’s efficiency gains. To thoroughly address your question, we conducted additional experiments under multiple scenarios and resolutions. Below, we show the kernel benchmark results of applying STA to a (29, 45, 76) latent on RTX 4090 by padding to (30, 48, 80):
> | Methods   | Implementation | Config        | Sparsity | TFLOPS | Latency(ms) | MFU    | Kernel Efficiency | Speedup |
> |-----------|----------------|---------------|----------|--------|-------------|--------|-------------------|---------|
> | Full Attn | FA2            | -             | 0.00%    | 141.22 | 825.46      | 51.84% | 100.00%           | 1.00×   |
> | Full Attn | Flex Attn      | -             | 0.00%    | 141.22 | 873.78      | 48.98% | 94.47%            | 0.94×   |
> | STA       | Flex Attn      | w=(29,24,24)| 85.00%   | 21.18  | 149.77      | 42.86% | 82.68%            | 5.51×   |
> | STA       | Flex Attn      | w=(18,24,24)| 89.55%   | 14.76  | 89.76       | 49.83% | 96.12%            | 9.20×   |
>
>
> Our evaluations suggest this padding overhead is negligible compared to the substantial latency reduction achieved by STA.
> We also present end-to-end results applying STA to Wan 2.1 at a different resolution (93 frames × 1024 × 1024) in a training-free setup, further demonstrating STA is applicable to varying input dimensions:
>  | Steps | Attn  Sparsity                  | SSIM ↑ | PSNR ↑ | Latency | Speedup |
> |------------|-----------------------------------|--------|--------|---------|---------|
> | 50 steps full attn|   0.0%    |  –|   –   | 1996s    | 1.00×   |
> | 18 steps full + 32 steps sta|   50.31%    | 77.41 | 20.16  | 1322s    | 1.51×   |
>
>
> We will explicitly clarify these practical constraints and solutions in the revised manuscript to better inform readers.
> We hope the additional experimental results address your concern and clarify the constraints involved. If our response resolves your reservations, we would be grateful if you would consider raising your score. We also welcome any further questions or suggestions you may have.

---

### Decision · Program_Chairs · 2025-05-01

**Decision:**

Accept (poster)

**Comment:**

The paper introduces Sliding Tile Attention (STA), a novel attention mechanism to accelerate video generation in Diffusion Transformers (DiTs) by leveraging localized 3D attention patterns. STA achieves significant end-to-end speedups (1.36–3.53×) with minimal or no quality loss, supported by comprehensive experiments on efficiency and quality metrics. The method is hardware-efficient, with a tile-by-tile sliding window design, and shows generalizability across models like Hunyuan and Wan 2.1.

Strengths
* significant contribution: STA addresses a critical bottleneck in video generation, reducing attention computation costs with a novel, hardware-aware approach.
* robust evaluation: extensive experiments validate efficiency (MFU, latency) and quality (SSIM, PSNR, human evaluations), with clear comparisons to baselines like FlashAttention-3.
* generalizability: additional results on Wan 2.1 and FLUX demonstrate applicability beyond the primary model, addressing reviewer concerns.
* theoretical soundness: Theorems and formulations are clear and correct, enhancing understanding of efficiency gains.

Weaknesses
* limited model diversity: while generalizability is shown, some reviewers noted the need for broader testing across architectures like CogVideoX or Mochi.
* supplementary material: Lack of demo videos or initial supplementary material was a concern, though addressed in rebuttals with a provided link.
* training details: some reviewers requested more specifics on STA training, which authors partially addressed by referencing the appendix and promising open-source code.
* input constraints: STA’s reliance on tile-aligned dimensions requires padding or cropping, which introduces minor overhead, though mitigated in practice.

Overall, the paper presents a significant, well-evaluated contribution to video generation efficiency with broad applicability. The rebuttal effectively resolves reviewer concerns, and the promise of open-source code ensures reproducibility.